# Boosting Deductive Reasoning with Step Signals In RLHF

## Abstract

Logical reasoning is a crucial task for Large Language Models (LLMs), enabling them to tackle complex problems. Among reasoning tasks, multi-step reasoning poses a particular challenge. Grounded in the theory of formal logic, we have developed an automated method, Multi-step Deduction (MuseD), for deductive reasoning data. MuseD has allowed us to create training and testing datasets for multi-step reasoning. Our generation method enables control over the complexity of the generated instructions, facilitating training and evaluation of models across different difficulty levels. Through RLHF training, our training data has demonstrated significant improvements in logical capabilities for both in-domain of out-of-domain reasoning tasks. Additionally, we have conducted tests to assess the multi-step reasoning abilities of various models.

## 1 Introduction

Recent advancements in large language models (LLMs) (Ouyang et al., 2022; Bai et al., 2022) have yielded remarkable outcomes. Among the various capabilities of LLMs, reasoning stands out as one of crucial skills, serving as a foundational ability required for solving complex tasks. Numerous efforts (Sun et al., 2023) have been made to explore and enhance the reasoning capabilities of LLMs.

In this work, we focus on multi-step deductive reasoning tasks (Sun et al., 2023) within the realm of reasoning. Many previous works (Han et al., 2024; Saparov & He, 2023; Saparov et al., 2024) have been done of deductive reasoning data generation. However, most works concentrate on supervised fine-tuning (Sanh et al., 2022) or evaluation. Our work mainly concentrates on generation data for Reinforcement Learning from Human Feedback (RLHF) (Ouyang et al., 2022). Deductive reasoning tasks concentrate on deriving correct conclusions from given premises through rigorous and effective reasoning. Our attention is directed towards constructing high-quality data to improve the deductive reasoning abilities of models during the alignment phase. The inherent rigor of deductive reasoning tasks dictates that the corresponding prompts should not contain contradictory information or be incapable of leading to the correct answer. This presents our first challenge: how to obtain prompts with correct answers and no contradictions. Based on the proper conditions in prompts, we expect LLMs to employ multi-step reasoning to deduce the correct answer. Assessing the correctness of such a multi-step reasoning process constitutes our second issue, since we need accurate scores of responses to construct training data. Lastly, how to efficiently acquire a substantial amount of data for training is the third problem we need to consider.

To address these issues, we propose a generation scheme for multi-step deductive reasoning data, named Multi-step Deduction (MuseD). MuseD is a scalable approach from prompt creation to final evaluation. To sum up, we base our method MuseD on the syllogistic reasoning of deductive inference (Copi et al., 2016), employing a backward generation approach to obtain the conditions required for the prompt. This ensures that the conditions in the prompt can lead to the correct conclusion without any contradictory conditions. Moreover, by controlling the number of generated conditions, we can regulate the number of inference steps required for the prompt. Based on the prompts generated by this method, we can score the responses of LLMs step by step. That is, we provide an evaluation method that can assess whether an answer was correctly obtained through multi-step reasoning rather than merely being the correct answer. The data generation process is shown in Fig. 1.

Using MuseD, we synthesize partially ordered data for multi-step deductive reasoning and use this data for Reinforcement Learning from Human Feedback (RLHF) training. We achieve significant performance enhancements on both in-domain and out-of-domain reasoning datasets, validating the effectiveness of our synthesized data. Further experiments demonstrate that step scoring is crucial for improving the model's performance in RLHF and that positive rewards for correct answers are the primary motivator for model learning.

Additionally, we utilize our method to create a multi-step deductive reasoning evaluation set, also named MuseD. Compared to previous evaluation sets, it can provide insights into the model's performance changes under tasks with different numbers of reasoning steps and offer more granular process evaluations. In summary, our main contributions include: (1) proposing a data synthesis method MuseD based on multi-step deductive reasoning, (2) achieving improvements in deductive reasoning performance on RLHF with this method, and (3) presenting a multi-step deductive reasoning evaluation set that allows for multi-dimensional automatic scoring.

We will briefly introduce the logical concepts involved in our method in Sec. 3 and describe our data generation approach in Sec. 4. In Sec. 5, we will present the experimental results, and in Sec. 6, we will show the performances of some LLMs on our evaluation set.

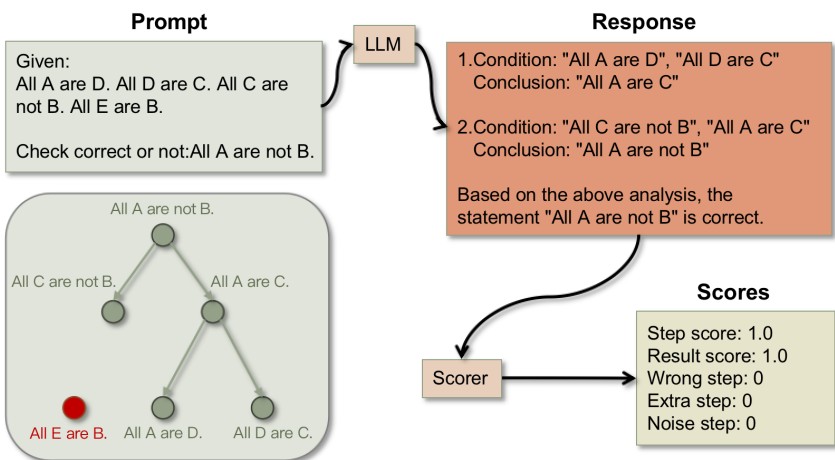

Figure 1: The logic training data pipline.

## 2 RELATED WORK

Many recent works concentrate on the reasoning ability of LLM. Sun et al. (2023) give a survey on LLM reasoning, where different kinds of reasoning are considered. Traditionally, reasoning are divided into deductive reasoning (Saparov & He, 2023), inductive reasoning (Wang et al.) and abductive reasoning (Bhagavatula et al.).

Our work concentrates on deductive reasoning, where valid reasoning process are focused. Many efforts have been done on this tasks to provide high-quality datasets. Tafjord et al. (2021) give a pipline to generate a synthetic data based on rules. Liu et al. (2021) extend human logical testing to LLM dataset. Han et al. (2024) concentrate on natural logical data that are created by experts. Saparov & He (2023) and Saparov et al. (2024) use deduction rules to generate synthetic data and provide datasets. Our work takes inspirations from all these previous works and we combine syllogism rules to develop synthetic data. Specially, we provide not only a dataset, but also a pipline that can give scores on LLM responses. Therefore, our method is suitable for on-policy training such as reward model and PPO training.

There are also many works that improve reasoning ability from algorithmic aspect. Wei et al. (2022) use Chain-of-Thought (CoT) method to improve reasoning ability, while Wang & Zhou (2024) try some decoding method. Havrilla et al. and Pang et al. (2024) explore improving reasoning ability of LLM through RLHF and iterative DPO respectively. Kumar et al. (2024) attempt to incentive the self-correction ability of LLM. Our method mainly concentrates on RLHF method with our

generated dataset. We show that data generation process has large performance effect on RLHF method.

## 3 PRELIMINARY

### 3.1 PROPOSITION

The proposition provides an object that we can judge to be true or false. Logical reasoning focuses on the process of deriving a conclusion from one or more premises. These premises and conclusions often appear in the form of propositions. In natural language, the expression of a proposition may be complex and diverse and it is hard to deal with such flexible data. In this article, we mainly focus on the syllogistic deduction in formal logic, which has relatively simple structure. Therefore, we mainly pay attention to the categorical propositions that judge the relationship between the subject and predicate. In categorical propositions, we judge the relationship between members of one category (the subject of the proposition) and another category (the object of the proposition). Aristotle has established four standard forms of categorical propositions, which we generally write as A, E, I, O (Aristotle, 350 BCE; Copi et al., 2016). Below we use S as the subject and P as the predicate, and we give the formats of the four forms of propositions: (1) form *A*: All S are P; (2) form *E*: No S are P; (3) Form *I*: There is one S that is P; (4) form *O*: There is one S that is not P.

The four propositional structures presented here provide fundamental statements for judging the relationship between two entity categories. Such structures are independent of the actual semantics of the entities, allowing us to focus more easily on the formal relationships. For large-scale model data production, based on such formal structures, we can effectively carry out data production.

### 3.2 SYLLOGISM

Herein, we provide a succinct introduction to the categorical syllogism in formal logic (Copi et al., 2016). The data generation process in our subsequent work is predicated on these effective deductive reasoning rules.

The categorical syllogism is an ancient method of logical argumentation that employs deductive reasoning to study the inferential process of deriving a conclusion from two premises. In the syllogism, we refer to the subject of the conclusion as the minor term, and the predicate as the major term. The two premises of the syllogism each contain the major and minor terms, along with a term that connects the two premises, which we call the middle term. The premise containing the major term is referred to as the major premise, while the premise containing the minor term is called the minor premise. For instance, we may cite a classic example of a syllogism:

> "All men are mortal. Socrates is a man. Therefore, Socrates is mortal."

In this example, the major term is 'mortal', the minor term is 'Socrates', and the middle term is 'men'. Hence, "All men are mortal" is the major premise, and "Socrates is a man" is the minor premise.

The inference process of a categorical syllogism involves establishing a relationship between the major and minor terms using the middle term. In a valid categorical syllogism, if both the major and minor premises are true, then the conclusion must be true. Formal logic has provided several valid syllogistic forms. In our work, we utilize the 15 valid syllogistic forms[1] found in modern formal logic (Copi et al., 2016) to generate our logical data.

## 4 MUSED METHOD

Given our knowledge of valid syllogistic forms, we can in turn generate two premises from a conclusion by introducing a middle term. Conversely, in the actual deductive process, the relationship between the major and minor terms is established by eliminating the middle term. Therefore we can

---

[1]These valid syllogistic forms can be found in the table of all syllogisms in the wiki website: https://en.wikipedia.org/wiki/Syllogism. We only use results in solid line boxes.

generate prompts in different complexity using the premises generation method. Moreover, we can score the responses based on the rules of deductive reasoning, thereby obtaining densely rewarded data. Additionally, our dataset can also serve as an evaluation set, allowing for the assessment of a model's deductive reasoning capabilities at various levels of complexity.

In the following, we will introduce the entire data generation process in four parts: prompt generation, response generation, response scoring and preference-pair composition.

### 4.1 PROMPT GENERATION

The prompt for logical problems should be rigorous and reasonable. We need to ensure that the prompts we generate can lead to correct conclusions through multi-step reasoning. Therefore, we must guarantee correctness during the prompt generation process. Our approach to constructing prompts is backward-generated, starting from the conclusion to be proven and progressively increasing the complexity of the prompt based on valid formal rules of inference.

Our generation process consists of three steps. The first step is to create a deductive reasoning logic tree where each node on this tree is a proposition. The root node is the conclusion and the leaf nodes are the conditions in prompts. The generated tree only gives a logical structure where all terms are not filled. In the second step, we fill in the content for the missing entities in this tree. Finally, we combine conditions and conclusions to get prompts.

Specifically, we initially employ the syllogistic method of formal logic to generate a multi-step logic tree from the ultimate target conclusion. At the beginning of the tree generation process, we create a root node, which takes the form of one of the four propositions randomly sampled from the previous context. This root node is the final conclusion of this prompt. Throughout the process, we maintain a set of leaf nodes, which initially contains only the root node. As we progress, we iteratively pop a node from the set of leaf nodes, use it as a conclusion, sample a valid categorical syllogism pattern, and transform this conclusion node into two premise propositions by introducing a middle term. These two premise propositions are then added to the set of leaf nodes. Each such sampling process increases the number of leaf nodes by 1 and also necessitates an additional step in the multi-step reasoning from the leaf nodes back to the root node. Thus, by setting the number of times we expand the leaf nodes, we can control the complexity of the generated prompt. When we ultimately have $N + 1$ leaf nodes, we turn them to $N + 1$ propositions to construct the prompt. From these propositions, the final conclusion can be inferred through $N$ steps of reasoning. Note that during the generation process, we use placeholders rather than specific entities as terms to generate a logical framework.

Based on the aforementioned logical conditions, we can also randomly introduce some interfering conditions. We introduce extra conditions that are related to the existing terms but do not conflict with current propositions, serving as noisy conditions in the reasoning process.

After establishing the logical trees, we then filling placeholders with entities. The most straightforward yet challenging approach is to populate it with noun concepts that conform to the logical conditions. However, this method can easily lead models to take shortcuts (Zhang et al., 2023; Saparov & He, 2023) in concluding judgments rather than genuinely engaging in logical reasoning. For instance, given the premises that cats are mammals and mammals are animals, the model can easily arrive at this conclusion that cats are animals with common sense judgment rather than using deductive reasoning. This is what we aim to avoid.

In our work, we focus on the formal structure of logical reasoning itself. Therefore, we have chosen two forms of virtual entities: Greek letter names and virtual nouns. Greek letter names are directly sampled, such as "ALPHA" or "BETA", to substitute for entities. Virtual nouns consist of 4-14 English letters and are added to our virtual noun database after confirming they are not actual words. We then sample the required quantity from this database to fill the prompts.

Finally, we transform the conditions and conclusion with added entities into a prompt. For the conditions, we concatenate them directly as the given premises. Regarding the conclusion, we have two questioning formats: proof and judgment. For proof questions, we directly ask the model to prove the conclusion. For judgment questions, we reverse the conclusion to its negation with a probability of 0.5 and ask the model to judge whether the given proposition is correct. The templates for proof and judgement are given in Appendix F.

## 4.2 RESPONSE GENERATION

Based on the generated prompts, we can directly access the model to obtain responses. We refer to this method as "natural" response generation. This approach yields the model's reasoning process expressed in natural language; however, such responses are not easily evaluated. We also incorporate instructions to generate responses in a specified JSON format within the prompt, allowing for structured answers that can be conveniently scored. We term this approach as "formatted" response generation.

The natural method applies solely to the prompt itself, without additional requirements for the format and style of the response. We believe this is more consistent with the model's output during user interaction and aligns with the distribution of natural language. However, such natural language is relatively challenging to handle during scoring, necessitating additional operations in the evaluation phase to achieve better scoring outcomes.

The formatted generation method, on the other hand, introduces a fixed format for responses, resulting in outputs that are easier to score. Nevertheless, the significant difference between such structured expressions and natural language may impact the learning process during model alignment. The added format is provided in a few-shot manner, with details outlined in Appendix G.

We sample responses to the generated prompts using both methods. In our experiments, we utilize the Llama3 8b model and evaluate the model's responses under these two approaches.

## 4.3 RESPONSE EVALUATION

The construction of formal logic problems are generated from the root node to the leaf node with the syllogism approach. Consequently, the procedure of proof or judgement constitutes the process of initiating from the leaf nodse and to arrive at the proposition of the root node.

We assess the responses of Large Language Models (LLM) on formal logic reasoning problems in multiple aspects:

- **Step score**: Calculate the how much correct step the response reaches. We mainly calculate this score by counting the eliminated middle terms, details given in Sec. 4.3.1.

- **Result score**: Evaluate whether the response reaches the correct conclusion. A score of 0/1 is assigned. For proof-type questions, if the proposition to be proved emerges in the reasoning process, a score of 1 point is awarded. For judgment-type questions, the score is directly allocated based on the correctness of the judgment.

- **Intent score**: Evaluate whether the generated formatted response is valid JSON string. If valid, a score of 1 is given; otherwise, a score of 0 is assigned.

- **Wrong step**: Count the number of wrong reason steps. The scores equals the number of wrong steps in the propositions within the responses.

- **Noise step**: There might be propositions in the responses that we cannot check its correctness or irrelevant to the reasoning process. We count the numbers of these steps as the noise step score.

- **Extra step**: For correct steps in the responses, we count the number of repeated steps.

Step score, Result score, and Intent score are positive indicators. The higher the value, the better the effect of the LLM. Wrong step score, Noise step score, and Extra step score are negative indicators. The lower the value, the better the effect of the model.

### 4.3.1 STEP SCORE CALCULATION

For multi-step reasoning processes, we aim to obtain accurate process scores to evaluate the quality of different answers. It should be noted that the reason process from premises to the final conclusion is not unique. For example, consider conditions: (1) A is B, (2) B is C, and (3) C is D, and we want to conclude that (6) A is D. Using syllogistic reasoning, we can first derive (4) A is C from (1) and (2), and then combine condition (3) to reach (6). Alternatively, we can derive (5) B is D from (2) and (3), and then combine condition (1) to reach the conclusion (6). Both methods are correct and should receive equally high step scores.

Recall that the process of expanding logic tree is to add middle terms to get conditions. Hence reasoning process towards the final conclusion is essentially the process of eliminating these middle terms. In the above example, there are two paths to conduct inference. The first path eliminating B first and then C, while the second path eliminating C first and then B.

Based on observation above, we can enumerate valid middle terms in the logical tree, i.e. term B and C in the above example. We can check the reasoning process of each response and count the number of middle terms that are eliminated. Repeated elimination would not be counted. We calculate the rate of the eliminated middle terms and the number of all middle terms to get the step score.

### 4.4 PREFERENCE-PAIR COMPOSITION

We need to construct preference data to train the reward model (RM), based on the scored data above. Different from Outcome Reward Model (ORM) (Lightman et al.), we have multiple scoring dimensions to construct pairs. Therefore our RM can learn to provide dense reward signals to help PPO training. On the other hand, we don't train RM in the Process Reward Model (PRM) pattern Lightman et al.; Wang et al. (2023), where rewards are assigned to each step, thus we can train RM with other datasets to ensure the general ability of RM.

Since each data point has multiple scoring dimensions, we can construct dataset with different composition methods, ultimately affecting the learning outcome of the model. Here, we experiment with three methods for generating data pairs:

- **P**: uses only positive scores (i.e., the step score and the result score) to construct pairs. For each pair, the chosen response must have a higher step score than the rejected one, while the result score of the chosen cannot be lower than that of the rejected one.

- **PN**: uses both positive and negative scores to construct pairs. We use all the scores above. For wrong, noise and extra scores, we use the opposite number to turn the negative signal into positive. For each pair, each score of the chosen response cannot be lower that that of the rejected one and the chosen should has at least one score that is higher.

- **R**: uses only result score as the pair standard. That is, the chosen response should has result score 1 while the rejected one has result score 0.

One important concern is that whether negative signals should be used. Intuitively, since it should be easier not to do wrong than to do right, negative signals may be easier to learn than positive ones. We compare method **P** and **PN** to see whether the choice of signals matters much. We also conduct method **R** to show the importance of step scores, since **R** only provides sparse final result signal, as usually used on ORM training.

## 5 EXPERIMENTS

### 5.1 EVALUATION SETS

We use three kinds of evaluation sets to test the performance of models. The in-domain set MuseD is generated and evaluated by our proposed method. The out-of-domain sets are open source logical sets to show whether our model indeed improve its deductive logical ability on prompts out of distribution. We finally choose some general open source datasets to test whether the general abilities are influenced. We set the sampling temperature to be 0.3 for MuseD, 0.001 for out-of-domain datasets.

- **In-domain set**: MuseD dataset with 2000 prompts, which is introduced in Sec. 6.

- **Out-of-domain sets**: For the open source logical verification set, we choose **PrOntoQA** (Saparov & He, 2023), **ProofWriter** (Tafjord et al., 2021), **LogicalDeduction** (Pan et al., 2023), **FO-LIO** (Han et al., 2024), **AR-LSAT** (Zhong et al., 2021). We use the CoT (Chain of Thought) (Wei et al., 2022) method, and CoT prompt of each dataset from (Xu et al., 2024). The evaluation metric is accuracy, which measures the correctness of multiple-choice questions.

- **General ability sets** We follow the default implementation setting of LM Eval Harness (Gao et al., 2024) and set the temperature hyperparameter as 0. For general ability evaluation, we select the

task evaluation in Sec. D and follow the default implementation setting of LM Eval Harness with temperature hyperparameter as 0 and report 0-shot accuracy.

## 5.2 REWARD MODEL

The reward models (RM) are trained following the standard process of InstructGPT (Ouyang et al., 2022). We choose the Llama3 (Dubey et al., 2024) 8B pretrain model as our base model. If not specified, we keep the hyper-parameters of RMs fixed. More specifically, we set the batchsize to be 384 and the maximum learning rate to be 5e-6.

We train RMs with different preference datasets. We mainly to evaluate the performance of our constructed logic dataset. We choose UltraFeedback (UF) (Cui et al., 2023) as our base dataset to avoid ability decline in other fields, we use the fine-grained scores to get around UF 27w pairs. For our logic dataset, we compare the performances of the natural and formatted response generation methods. We also test the P, PN and R pair construction methods. All our RMs are listed below. For convenience, we use **Na** to denote natural responses and **Fo** to denote formatted responses.

- **RM-UF**: uses only 27w UF data.

- **RM-Na-P**: uses 15k natural P-pair logic data. UF data is used.

- **RM-Fo-P**: uses 15k formatted P-pair logic data. UF data is used.

- **RM-Mix-P**: uses 16k logic data, half from the logic data in RM-Na-P and half from the data in RM-Fo-P. UF data is used.

- **RM-Na-PN**: uses 11k natural PN-pair logic data. UF data is used.

- **RM-Na-R**: uses 8k natural R-pair logic data. UF data is used.

- **RM-NaO-P**: uses only 15k natural P-pair logic data. We turn the batchsize of this RM to be 96 to get more steps.

Notice that we use the same natural responses to construct P, PN and R pairs and different rules result in different size of dataset. For Fo-P data, we downsample them to the same size of Na-P data. We put the validation accuracy and RewardBench (Lambert et al., 2024) scores of these RMs in the Appendix C. Roughly speaking, each RM gains high validation accuracy on its trained datasets, and the reasoning scores in RewardBench are not directly consistent with our experimental results in Sec. 5, possibly due to the distribution mismatch.

## 5.3 PPO TRAINING

### 5.3.1 SETTINGS

We train PPO models using above RMs. The base model for PPO training is Llama3 8B Instruct model. The prompts used for training are the same prompts used for the corresponding RM. That is, the prompts used for one RM would be used for the PPO model that uses this RM. The hyperparameters for training PPO are fixed for all experiments. The experience batch size is 160 and the micro batch size is 20. The sampling temperature for training is 1.0. We train for only one epoch. For convenience, we denote each PPO model with its corresponding RM as its subscript. We remove the $RM-$ in the subscript for briefly. That is, we use $PPO_A$ to denote the PPO trained with RM-A.

Further, we compare the performance difference if we change the training prompts of PPO. We choose RM-Na-P as our baseline RM and use $PPO_{Na-P}$ as our baseline PPO model. We test the impact of using extra in-domain prompts. For $PPO_{Na-P}$, we use 6587 prompts trained for RM-Na-P. Then construct more prompts with our pipline MuseD to increase the logical prompts to reach 20000. We combine all these prompts with UF prompts to train one epoch over PPO training, and we get $PPO_{Na-P-All}$. We also sample 6587 prompts from the 20000 prompts to train PPO and we get $PPO_{Na-P-Replace}$. Finally, we test the effect of curriculum learning during PPO training. That is, we sort the logical prompts used in $PPO_{Na-P}$ with their levels. We train them from easy to hard and we get $PPO_{Na-P-Cur}$.

### 5.3.2 PERFORMANCES

We compare the performances of all PPO models on the three kinds of evaluation sets mentioned in Sec. 5.1. The performances of all models on our MuseD dataset, i.e. the in-domain dataset, are shown in Table 1. Table 2 gives comparison of models on the out-of-domain datasets. The results for general ability of models are shown in Appendix A. Compared with $PPO_{UF}$, the involvement of our data do not seem to cause degradation on other abilities.

Comparing to $PPO_{Na-P}$, we can see that $PPO_{NaO-P}$ shows bad performance on all scores, indicating the necessity of using a general dataset. Compared to Llama3 8B Instruct, the UF dataset only don't increase the step and result scores much, while on out-of-domain datasets, $PPO_{UF}$ has a obvious improvement. We check the cases and find that the bad performance on instruction following leads to the low scores of Llama3 instruct model. It can be seen from the results on AR-LSAT, which is a 5-choice dataset. A random strategy would leads to a score around 0.2 while Llama3-8b-Instruct gets a score near 0.

To show the effect of training with our dataset, we compare $PPO_{UF}$ and $PPO_{Na-P}$. The logical data yields a significant gain in logical effects. Compared to $PPO_{UF}$, $PPO_{Na-P}$ increases the step and result scores of MuseD both by 12 percentage points, and reduces the average wrong step significantly. The noise step and extra step indicators are basically the same. More results on level-wise comparison are given in Appendix B. On the public evaluation datasets, there is significant growth on ProntoQA, ProofWriter, and LogicalDeduction. Specifically, there are 14-16% improvements on ProntoQA and LogicalDeduction, which have quite similar deductive tasks as MuseD dataset. There is a slightly decrease on AR-LAST, a five-choice task dataset (refer to Appendix E.1). Although both PPO models get a reward similar to a random strategy, $PPO_{Na-P}$ sometimes refused to pick any answer when it thinks all options are wrong.

| Model | step score | result score | intent score | wrong step count | noise step count | extra step count |
|---|---|---|---|---|---|---|
| Llama3-8B-Instruct | 0.3485 | 0.5715 | 0.9495 | 1.685 | 0.7815 | 0.054 |
| $PPO_{UF}$ | 0.3135 | 0.5725 | 0.9905 | 1.252 | 0.438 | 0.0305 |
| $PPO_{Na-P}$ | 0.4383 | 0.6975 | 0.9925 | 0.85 | 0.3865 | 0.036 |
| $PPO_{NaO-P}$ | 0.4117 | 0.4920 | 0.8695 | 2.3715 | 1.0515 | 0.5585 |
| $PPO_{Na-P-All}$ | **0.4993** | **0.7585** | 0.9905 | 0.6645 | 0.3595 | 0.0805 |
| $PPO_{Na-P-Replace}$ | 0.431 | 0.676 | 0.985 | 0.842 | 0.425 | 0.05 |
| $PPO_{Na-P-Cur}$ | 0.4591 | 0.6855 | 0.991 | 0.9515 | 0.455 | 0.0655 |
| $PPO_{Na-R}$ | 0.3472 | 0.6605 | **0.9975** | 0.8115 | 0.257 | 0.018 |
| $PPO_{Na-PN}$ | 0.3649 | 0.6675 | 0.99 | **0.672** | **0.2155** | **0.0125** |
| $PPO_{Fo-P}$ | 0.3744 | 0.6205 | 0.9725 | 1.0945 | 0.367 | 0.053 |
| $PPO_{Mix-P}$ | 0.4455 | 0.693 | 0.993 | 1.135 | 0.5615 | 0.0645 |

Table 1: Performances of Lamma3 8B models trained with different RMs on MuseD dataset.

| Model | ProntoQA | ProofWriter | LogicalDeduction | FOLIO | AR-LSAT | Average |
|---|---|---|---|---|---|---|
| Llama3-8B-Instruct | 0.616 | 0.1367 | 0.0567 | 0.1813 | 0.0043 | 0.199 |
| $PPO_{UF}$ | 0.728 | 0.37 | 0.3267 | 0.4118 | 0.2165 | 0.411 |
| $PPO_{Na-P}$ | **0.868** | 0.4567 | **0.4833** | 0.451 | 0.1948 | **0.4908** |
| $PPO_{NaO-P}$ | 0.352 | 0.2467 | 0.0967 | 0.2451 | 0.0519 | 0.1985 |
| $PPO_{Na-P-All}$ | 0.672 | 0.4517 | 0.32 | **0.4804** | 0.1905 | 0.4229 |
| $PPO_{Na-P-Replace}$ | 0.818 | **0.4967** | 0.3933 | 0.4803 | 0.2035 | 0.4784 |
| $PPO_{Na-P-Cur}$ | 0.762 | 0.4717 | 0.4066 | 0.4706 | 0.1861 | 0.4594 |
| $PPO_{Na-R}$ | 0.798 | 0.4017 | 0.3867 | 0.4069 | **0.2294** | 0.4445 |
| $PPO_{Na-PN}$ | 0.816 | 0.445 | 0.2667 | 0.402 | 0.1948 | 0.4249 |
| $PPO_{Fo-P}$ | 0.804 | 0.4033 | 0.44 | 0.4461 | 0.1905 | 0.4568 |
| $PPO_{Mix-P}$ | 0.806 | 0.4367 | 0.43 | 0.4265 | 0.1991 | 0.4597 |

Table 2: Performances of Lamma3 8B models trained with different RMs on out-domain logical evaluation sets.

### 5.3.3 PREFERENCE PAIR COMPOSITION

We compare the performance of models trained with PPO across different preference pair generation methods which are presented in Section 4.4. We compare the performances of $\text{PPO}_{\text{Na}-\text{P}}$, $\text{PPO}_{\text{Na}-\text{PN}}$ and $\text{PPO}_{\text{Na}-\text{R}}$.

**Process signals can significantly improve the effect.** RM-Na-P uses preference data incorporing process signals while RM-Na-R uses data constructed with result score only. The step score exhibits significant growth on the MuseD evaluation set. On the public datasets, $\text{PPO}_{\text{Na}-\text{P}}$ is notably superior to $\text{PPO}_{\text{Na}-\text{R}}$. There are about 10-point improvement LogicalDeduction, 7-point imporvement on ProntoQA , and 5-point improvements on both ProofWriter and FOLIO .

**Negative signals will cause degradation in effectiveness.** Introducing negative signals for the construction of preference data in the reward model will notably diminish the effectiveness of the PPO model. $\text{PPO}_{\text{Na}-\text{PN}}$ further incorporates negative indicators on the foundation of $\text{PPO}_{\text{Na}-\text{P}}$. There is indeed a reduction in the three dimensions of negative indicators (wrong step count, noise step count, and extra step count) in MuseD. However, it significantly deteriorates in the positive indicator step score. Concurrently, on the public logical verification set, ProntoQA, LogicalDeduction, and FOLIO exhibit a significant decline.

### 5.3.4 NATURAL RESPONSE VS. FORMATTED RESPONSE

We also investigate the influence of data format of prompts. We choose Na (natural responses) and Fo (formatted responses). We train three PPO models with distinct data formats: $\text{PPO}_{\text{Na}-\text{P}}$ (utilizing Na), $\text{PPO}_{\text{Fo}-\text{P}}$ (employing Fo), and $\text{PPO}_{\text{Mix}-\text{P}}$ (mixing Na and Fo in a 1:1 ratio).

**Natural format performs better.** $\text{PPO}_{\text{Na}-\text{P}}$ is notably superior to $\text{PPO}_{\text{Fo}-\text{P}}$ and $\text{PPO}_{\text{Mix}-\text{P}}$. On MuseD dataset, $\text{PPO}_{\text{Na}-\text{P}}$ exhibits significant superiority over $\text{PPO}_{\text{Fo}-\text{P}}$, boasting a 6-7 point advantage in step score and result score. In comparison to $\text{PPO}_{\text{Mix}-\text{P}}$ version, $\text{PPO}_{\text{Na}-\text{P}}$ version is generally on an equal footing in positive indicators. However, regarding negative indicators such as incorrect step count, noise step count, and extra step count, $\text{PPO}_{\text{Na}-\text{P}}$ version is markedly better than $\text{PPO}_{\text{Mix}-\text{P}}$ version. On the public test logic test collection, $\text{PPO}_{\text{Na}-\text{P}}$ version surpasses $\text{PPO}_{\text{Fo}-\text{P}}$ and $\text{PPO}_{\text{Mix}-\text{P}}$ version in the ProntoQA, ProofWriter, LogicalDeduction, and FOLIO datasets. The result shows that natural format, which is more closely to the natural language, is more effective for RLHF training.

### 5.3.5 VARIANTS OF LOGICAL PROMPTS IN PPO TRAINING

We also experiment on the impact of using additional prompts on top of the prompts covered by RM training. We compare the performances among $\text{PPO}_{\text{Na}-\text{P}}$, $\text{PPO}_{\text{Na}-\text{P}-\text{All}}$ and $\text{PPO}_{\text{Na}-\text{P}-\text{Replace}}$.

Compare $\text{PPO}_{\text{Na}-\text{P}-\text{All}}$ with $\text{PPO}_{\text{Na}-\text{P}}$. It can be observed that by adding in-domain prompts, the effect on the MuseD set is significantly increased, while there is a significant drop on the out-of-domain data set. It seems that RM-Na-P has the ability to get relatively correct scores for these in-domain prompts, which helps PPO model to improve. However, a too high data proportion will cause overfittin on MuseD data and reduce the effect on non-identically distributed datasets.

Compare the two experiments of $\text{PPO}_{\text{Na}-\text{P}-\text{Replace}}$ and $\text{PPO}_{\text{Na}-\text{P}}$. In the MuseD set, $\text{PPO}_{\text{Na}-\text{P}}$ exhibits superiority in all indicators to $\text{PPO}_{\text{Na}-\text{P}-\text{Replace}}$. On the publicly available logic test set, $\text{PPO}_{\text{Na}-\text{P}-\text{Replace}}$ shows worse performance on ProntoQA and LogicalDeduction but better performance on FOLIO and ProofWriter. On average, it decreased by approximately 1 point. Generally, using prompts trained with the reward model to train PPO yields better effects both on in-distribution prompts and on open-source data outside the distribution. The possible reason is that on the trained prompts, the reward model scores more accurately, thereby better guiding the training process of PPO. As for the better performance on FOLIO and ProofWriter, it might be caused by a bit over-fitting on the MuseD deductive format, which hurts the reasoning ability on tasks like FOLIO or ProofWriter a bit, since ProntoQA and LogicalDeduction have more similar formats to MuseD.

### 5.3.6 CURRICULUM LEARNING

We also conducte an experiment on curriculum learning as $\text{PPO}_{\text{Na}-\text{P}-\text{cur}}$ to study its impact on the effect. We uniformly incorporate data of varying levels, ranging from low to high, into the prompt of

PPO training. Ensure that with PPO steps training, the traversed logical data progresses from simple to complex.

It can be observed that in comparison to shuffling data, employing the mechanism of curriculum learning: on the MuseD set, the overall result is comparable. However, on the open-source test set, the performance drops. On ProntoQA and LogicalDeduction, it declines by 8 to 10 percentage points.

## 6 EVALUATIONS ON LLMS

Our constructed data can also be used as an evaluation dataset to test the performance of LLMs on multi-step deductive reasoning. We generate 2000 prompts to compose this evaluation sets, which we called MuseD. MuseD includes questions that multi-step deductive reasoning is needed. The reason steps range from 1 to 10.

We test the effects of some models on MuseD and the results are shown in Table 3. Obviously, GPT-4-o1-Preview shows best performance. Qwen2.5-72B-Instruct also reaches high scores on step and result scores.

| Model | step score | result score | intent score | wrong step count | noise step count | extra step count |
|---|---|---|---|---|---|---|
| GPT-4 | 0.7306 | 0.8250 | 0.9995 | 1.1015 | 0.4785 | **0.0125** |
| GPT-4o | 0.8085 | 0.8320 | 0.9890 | 0.6940 | 0.7490 | 0.1015 |
| GPT-4-o1-mini | 0.4514 | 0.6635 | **1.0000** | 1.3570 | 0.6550 | 0.0290 |
| GPT-4-o1-preview | **0.8516** | **0.8895** | 0.9785 | **0.2785** | **0.0380** | 0.0800 |
| Qwen2.5-72B-Instruct | 0.8236 | 0.8844 | 0.9995 | 0.6386 | 0.5796 | 0.0581 |
| Qwen2-72B-Instruct | 0.7316 | 0.8050 | 0.9980 | 1.2130 | 0.9310 | 0.0675 |
| Llama3.1-72B-Instruct | 0.7543 | 0.8370 | 0.9920 | 1.3005 | 0.8135 | 0.1020 |

Table 3: Performances of Different Models on MuseD.

Using our MuseD evaluation set, we can see the detailed performances of LLMs. Besides the positive and negative scores above, we show the performance of LLMs over different levels and noise counts, as shown in Appendix H.

## 7 CONCLUSION

In this work, we propose a multi-step deductive data generation pipline, MuseD, including prompt generation, response scoring and pair composition. MuseD can construct prompts with controllable complexity and check the step scores of responses. We validate the effect of our generated logical data on Lamma3 8B instruct model with RLHF method. The result shows that our data can lead to significant improvement on in-domain of out-of-domain deductive reasoning tasks. Further, we show that natural format and positive step signals are important for RLHF. Finally, we use our pipline to generate an evaluation dataset, also named MuseD, to evaluate the performance of current LLMs.

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
