.** $PPO_{Na-P}$ is notably superior to $PPO_{Fo-P}$ and $PPO_{Mix-P}$. On MuseD dataset, $PPO_{Na-P}$ exhibits significant superiority over $PPO_{Fo-P}$, boasting a 6-7 point advantage in step score and result score. In comparison to $PPO_{Mix-P}$ version, $PPO_{Na-P}$ version is generally on an equal footing in positive indicators. However, regarding negative indicators such as incorrect step count, noise step count, and extra step count, $PPO_{Na-P}$ version is markedly better than $PPO_{Mix-P}$ version. On the public test logic test collection, $PPO_{Na-P}$ version surpasses $PPO_{Fo-P}$ and $PPO_{Mix-P}$ version in the ProntoQA, ProofWriter, LogicalDeduction, and FOLIO datasets. The result shows that natural format, which is more closely to the natural language, is more effective for RLHF training.

### 5.3.5 VARIANTS OF LOGICAL PROMPTS IN PPO TRAINING

We also experiment on the impact of using additional prompts on top of the prompts covered by RM training. We compare the performances among $PPO_{Na-P}$, $PPO_{Na-P-All}$ and $PPO_{Na-P-Replace}$.

Compare $PPO_{Na-P-All}$ with $PPO_{Na-P}$. It can be observed that by adding in-domain prompts, the effect on the MuseD set is significantly increased, while there is a significant drop on the out-of-domain data set. It seems that RM-Na-P has the ability to get relatively correct scores for these in-domain prompts, which helps PPO model to improve. However, a too high data proportion will cause overfittin on MuseD data and reduce the effect on non-identically distributed datasets.

Compare the two experiments of $PPO_{Na-P-Replace}$ and $PPO_{Na-P}$. In the MuseD set, $PPO_{Na-P}$ exhibits superiority in all indicators to $PPO_{Na-P-Replace}$. On the publicly available logic test set, $PPO_{Na-P-Replace}$ shows worse performance on ProntoQA and LogicalDeduction but better performance on FOLIO and ProofWriter. On average, it decreased by approximately 1 point. Generally, using prompts trained with the reward model to train PPO yields better effects both on in-distribution prompts and on open-source data outside the distribution. The possible reason is that on the trained prompts, the reward model scores more accurately, thereby better guiding the training process of PPO. As for the better performance on FOLIO and ProofWriter, it might be caused by a bit overfitting on the MuseD deductive format, which hurts the reasoning ability on tasks like FOLIO or ProofWriter a bit, since ProntoQA and LogicalDeduction have more similar formats to MuseD.

### 5.3.6 CURRICULUM LEARNING

We also conducte an experiment on curriculum learning as $PPO_{Na-P-cur}$ to study its impact on the effect. We uniformly incorporate data of varying levels, ranging from low to high, into the prompt of

PPO training. Ensure that with PPO steps training, the traversed logical data progresses from simple to complex.

It can be observed that in comparison to shuffling data, employing the mechanism of curriculum learning: on the MuseD set, the overall result is comparable. However, on the open-source test set, the performance drops. On ProntoQA and LogicalDeduction, it declines by 8 to 10 percentage points.

# 6 EVALUATIONS ON LLMS

Our constructed data can also be used as an evaluation dataset to test the performance of LLMs on multi-step deductive reasoning. We generate 2000 prompts to compose this evaluation sets, which we called MuseD. MuseD includes questions that multi-step deductive reasoning is needed. The reason steps range from 1 to 10.

We test the effects of some models on MuseD and the results are shown in Table 3. Obviously, GPT-4-o1-Preview shows best performance. Qwen2.5-72B-Instruct also reaches high scores on step and result scores.

| Model | step score | result score | intent score | wrong step count | noise step count | extra step count |
|---|---|---|---|---|---|---|
| GPT-4 | 0.7306 | 0.8250 | 0.9995 | 1.1015 | 0.4785 | **0.0125** |
| GPT-4o | 0.8085 | 0.8320 | 0.9890 | 0.6940 | 0.7490 | 0.1015 |
| GPT-4-o1-mini | 0.4514 | 0.6635 | **1.0000** | 1.3570 | 0.6550 | 0.0290 |
| GPT-4-o1-preview | **0.8516** | **0.8895** | 0.9785 | **0.2785** | **0.0380** | 0.0800 |
| Qwen2.5-72B-Instruct | 0.8236 | 0.8844 | 0.9995 | 0.6386 | 0.5796 | 0.0581 |
| Qwen2-72B-Instruct | 0.7316 | 0.8050 | 0.9980 | 1.2130 | 0.9310 | 0.0675 |
| Llama3.1-72B-Instruct | 0.7543 | 0.8370 | 0.9920 | 1.3005 | 0.8135 | 0.1020 |

Table 3: Performances of Different Models on MuseD.

Using our MuseD evaluation set, we can see the detailed performances of LLMs. Besides the positive and negative scores above, we show the performance of LLMs over different levels and noise counts, as shown in Appendix H.

# 7 CONCLUSION

In this work, we propose a multi-step deductive data generation pipline, MuseD, including prompt generation, response scoring and pair composition. MuseD can construct prompts with controllable complexity and check the step scores of responses. We validate the effect of our generated logical data on Lamma3 8B instruct model with RLHF method. The result shows that our data can lead to significant improvement on in-domain of out-of-domain deductive reasoning tasks. Further, we show that natural format and positive step signals are important for RLHF. Finally, we use our pipline to generate an evaluation dataset, also named MuseD, to evaluate the performance of current LLMs.

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

## A  GENERAL ABILITY SETS EVALUATION

Here, we present the evaluation results of various models on general datasets. We mainly compare PPO models using our dataset with $PPO_{UF}$. For most datasets, there is little difference in performance among the models. On the gsm8k dataset, there is a significant variation in model performance, with some models showing improvement while others experiencing a slight decline.

| Model | hellaswag | lambada_openai | mmlu | gsm8k | openbookqa | triviaqa | arc_easy | arc_challenge | truthfulqa | average |
|---|---|---|---|---|---|---|---|---|---|---|
| Llama3-8B-Instruct | 57.7 | 71.9 | 63.8 | 32.6 | 34.0 | 51.1 | 81.6 | 52.6 | 43.9 | 54.4 |
| $PPO_{UF}$ | 57.7 | 71.8 | 64.1 | 37.7 | 34.2 | 44.4 | 81.4 | 53.8 | 47.2 | 54.7 |
| $PPO_{Na-P}$ | 57.9 | 71.8 | 64.1 | 34.6 | 34.4 | 47.6 | 81.9 | 54.1 | 46.8 | 54.8 |
| $PPO_{Fo-P}$ | 58.0 | 71.7 | 64.1 | 39.0 | 34.2 | 47.3 | 82.2 | 54.4 | 47.3 | 55.4 |
| $PPO_{Mix-P}$ | 58.0 | 71.8 | 64.1 | 43.4 | 34.4 | 43.6 | 81.8 | 53.6 | 46.8 | 55.3 |
| $PPO_{NaO-P}$ | 57.6 | 71.9 | 64.1 | 33.1 | 34.4 | 52.3 | 81.6 | 52.6 | 42.9 | 54.5 |
| $PPO_{Na-R}$ | 57.9 | 71.9 | 64.1 | 41.4 | 34.0 | 45.6 | 81.7 | 54.0 | 47.4 | 55.3 |
| $PPO_{Na-PN}$ | 57.8 | 71.8 | 64.1 | 36.6 | 34.4 | 45.4 | 81.6 | 53.9 | 47.0 | 54.7 |
| $PPO_{Na-P-All}$ | 58.0 | 71.8 | 64.1 | 40.2 | 34.0 | 44.6 | 81.7 | 53.9 | 47.2 | 55.1 |
| $PPO_{Na-P-Replace}$ | 57.9 | 71.8 | 64.0 | 38.2 | 34.0 | 45.7 | 81.6 | 53.7 | 46.8 | 54.9 |

Table 4: Performances of PPO Trained Model with different reward model on general ability evaluate sets.

## B  DETAIL COMPARISON

We give some detailed comparison between $PPO_{UF}$ and $PPO_{Na-P}$. We plot the performances of the two models over different levels, ranging from 1 to 10. Here the level is the necessary steps needed to solve the deductive task. The result is shown in Fig. **??**. It can be see that on step and result scores, $PPO_{Na-P}$ outperforms $PPO_{UF}$ on all levels. $PPO_{Na-P}$ also has a lower wrong step count, even through we do not use negative signals when constructing preference data. It can also be see that as the level be larger, performances of models become worse.

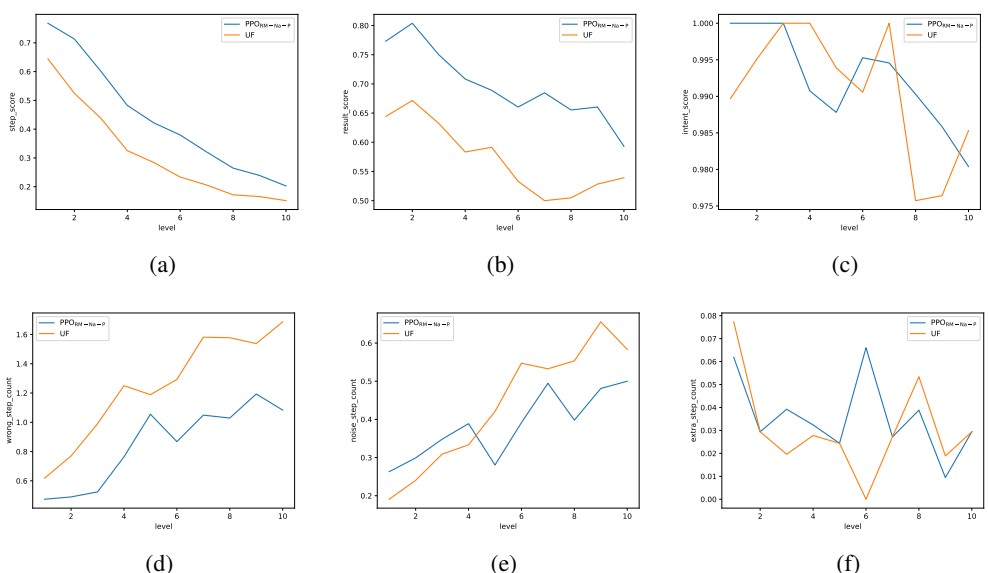

Figure 2: Performances of $PPO_{UF}$ and $PPO_{Na-P}$ over differnt levels.

## C  VALIDATION SET ACCURACY AND REWARDBENCH SCORES OF RMS

For each training dataset, we split a small set as validation set to test the accuracy of RMs. For logic and UF datasets, we split some prompts so that all RMs haven't been trained on these prompts. We use UF-2667 to denote the validation set of UF dataset with 2667 pairs. For each logic dataset with

A responses and B pair method, we denote its validation set with c pairs as "Logic-A-B-c". The results are given below.

| Validation set | RM-UF | RM-Na-P | RM-Fo-P | RM-Mix-P | RM-Na-PN | RM-Na-R | RM-NaO-P |
|---|---|---|---|---|---|---|---|
| UF-2667 | 0.8688 | 0.8635 | **0.8733** | 0.868 | 0.8695 | 0.8643 | 0.6832 |
| Logic-Na-P-1673 | 0.5415 | 0.7944 | 0.7286 | **0.7974** | 0.7155 | 0.6629 | 0.7603 |
| Logic-Fo-P-1314 | 0.6058 | 0.777 | **0.946** | 0.9125 | 0.7154 | 0.6705 | 0.7032 |
| Logic-Na-PN-1045 | 0.5898 | 0.7636 | 0.7368 | 0.7531 | **0.8411** | 0.6852 | 0.6871 |
| Logic-Na-R-729 | 0.5617 | 0.8944 | 0.7682 | 0.8971 | 0.882 | **0.9396** | 0.8203 |

Table 5: Accuracy of RMs over all validation sets

It can be see that most models has high accuracy on validation sets corresponding to their own training data. RM-UF has low accuracy on our logical data. However, we gain higher accuracy if we add UF dataset to our logical set, as shown by RM-Na-P and RM-NaO-P.

For each dataset, we test these RMs on RewardBench.

| RM models | Chat | Chat Hard | Reasoning | Safety | Average Accuracy |
|---|---|---|---|---|---|
| RM-UF | 0.9665 | 0.5811 | 0.7227 | 0.4972 | 0.6919 |
| RM-Na-P | 0.9749 | 0.6206 | 0.6971 | **0.5192** | 0.7029 |
| RM-Fo-P | **0.9832** | 0.6162 | 0.7105 | 0.5176 | 0.7069 |
| RM-Mix-P | 0.9693 | **0.6316** | **0.7362** | 0.4839 | 0.7052 |
| RM-Na-PN | **0.9832** | 0.6228 | 0.7286 | 0.4888 | 0.7059 |
| RM-Na-R | 0.9804 | **0.6316** | 0.7152 | 0.5046 | **0.708** |
| RM-NaO-P | 0.9134 | 0.4057 | 0.6318 | 0.4083 | 0.5898 |

Table 6: Accuracy of RMs over all validation sets

We can check the reasoning score of RewardBench on all the RMs with the final performances of PPOs. We find that this reasoning scores can not be a proper standard for RM chosen. This might be caused by the mismatch of both prompts and responses.

# D EVALUATION SETS DETAILS

We list all evaluation sets we used below.

- **lambda_openai**: Tasks designed to predict the endings of text passages, testing language prediction skills.Paperno et al. (2016)

- **mmlu**: Massive Multitask Language Understanding benchmark for broad domain language evaluation. Several variants are supported.Hendrycks et al. (2020)

- **gsm8k**: A benchmark of grade school math problems aimed at evaluating reasoning capabilities.Cobbe et al. (2021)

- **openbookqa**: Open-book question answering tasks that require external knowledge and reasoning.Mihaylov et al. (2018)

- **triviaqa**: A large-scale dataset for trivia question answering to test general knowledge.Joshi et al. (2017)

- **arc_easy**: Tasks involving complex reasoning over a diverse set of questions.Clark et al. (2018)

- **arc_challenge**: Tasks involving complex reasoning over a diverse set of questions.Clark et al. (2018)

- **truthfulqa**: A QA task aimed at evaluating the truthfulness and factual accuracy of model responses.Lin et al. (2022)

- **logicqa**: Logical reasoning tasks requiring advanced inference and deduction.Liu et al. (2021),

- **logicqa2**: Large-scale logical reasoning dataset adapted from the Chinese Civil Service Examination.Liu et al. (2023)

# E  EVALUATE SET TEMPLATE

## E.1  AR-LSAT

"id": "ar_lsat_201609_3-G_4_23",
"context": "There are exactly six computers—P, Q, R, S, T, and U—on a small network. Exactly one of those computers was infected by a virus from outside the network, and that virus was then transmitted between computers on the network. Each computer received the virus exactly once. The following pieces of information concerning the spread of the virus have been established: No computer transmitted the virus to more than two other computers on the network. S transmitted the virus to exactly one other computer on the network. The computer that transmitted the virus to R also transmitted it to S. Either R or T transmitted the virus to Q. Either T or U transmitted the virus to P.", "question": "If P is the only computer that transmitted the virus to two other computers on the network, which one of the following must be true?",
"options": [
"A) S transmitted the virus to T.",
"B) T transmitted the virus to P.",
"C) Q did not transmit the virus to any other computer on the network.",
"D) R did not transmit the virus to any other computer on the network.",
"E) U did not transmit the virus to any other computer on the network."
],
"answer": "C"

# F  TEMPLATES FOR PROOF AND JUDGEMENT PROMPTS

We use below templates to construct our prompts.

The proof template is

Given:
{Inference conditions.}
Prove: {conclusion}.

The judgement template is

We have:
{Inference conditions.}
Show {conclusion} is correct or not.

# G  TEMPLATES FOR FORMATTED RESPONSE GENERATION

To generate response with JSON format, we use few-shot learning to instruct model to generate. The template is given below. The template for proof problem is.

The template for judgement problem is given below. The template for proof problem is quite similar.

I will give you a few given conditions and you need to check whether a given conclusion is correct or not based on these conditions.
You need to list all the deductive process in a json style. For each step, you need to list:
* condition: the conditions you use to conduct deduction,
* conclusion: the conclusion you get,
* format conclusion: a dictionary which has below three terms:
** Subject: the subject of your conclusion, which should be an affirmed noun.

** Predication: the prediate of your conclusion, which should be an affirmed noun.
** type: which is one in ['A','E','I','O']. The type of one proposition with subject $S$ and predicate $P$:
*** Type 'A': 'all $S$ are $P$', or '$S$ is $P$'.
*** Type 'E': 'None of $S$ is $P$', or '$S$ is not $P$'.
*** Type 'I': 'There exists one $S$ that is $P$'.
*** Type 'O': 'There exists one $S$ that is not $P$'.
Finally you should give a 'result' if you are required to check whether the given conclusion is correct or not. If it is correct, return 'Correct'; otherwise, return 'Wrong'.

Your answer should be return with below format:
{'steps': [
'condition': ['xxx', 'xxx'],
'conclusion': ['xxx'],
'format_conclusion': {'Subject': 'xxx', 'Predicate': 'xxx', 'type', 'x'}
],[
'condition': ['xxx', 'xxx'],
'conclusion': ['xxx'],
'format_conclusion': {'Subject': 'xxx', 'Predicate': 'xxx', 'type', 'x'}
],[
'condition': ['xxx', 'xxx'],
'conclusion': ['xxx'],
'format_conclusion': {'Subject': 'xxx', 'Predicate': 'xxx', 'type', 'x'} ],
'result': 'xxx'
}

Examples:
{Examples}

##Input:
{Prompt}
##Output:

## H    DETAILED EVALUATION RESULTS

Here we give detailed evaluation results form LLMs on our MuseD dataset. We show the step scores, which we think is the most import score, over the deductive steps and the noise conditions.

| Levels | count | GPT-4 | GPT-4o | GPT-4-o1-mini | GPT-4-o1-preview | Qwen2-72B-Instruct | Qwen2.5-72B-Instruct | Llama3.1-72B-Instruct |
|---|---|---|---|---|---|---|---|---|
| 1 | 194 | 0.88 | **0.93** | 0.74 | **0.93** | 0.89 | 0.92 | 0.91 |
| 2 | 204 | 0.89 | 0.94 | 0.66 | **0.96** | 0.89 | 0.95 | 0.88 |
| 3 | 204 | 0.83 | 0.92 | 0.59 | **0.94** | 0.84 | **0.94** | 0.82 |
| 4 | 216 | 0.75 | 0.86 | 0.49 | **0.9** | 0.78 | 0.89 | 0.78 |
| 5 | 164 | 0.79 | 0.86 | 0.46 | **0.87** | 0.73 | **0.87** | 0.84 |
| 6 | 212 | 0.71 | 0.8 | 0.41 | **0.87** | 0.73 | 0.84 | 0.74 |
| 7 | 184 | 0.64 | 0.73 | 0.32 | 0.77 | 0.66 | **0.78** | 0.7 |
| 8 | 206 | 0.65 | 0.72 | 0.31 | **0.81** | 0.64 | 0.73 | 0.65 |
| 9 | 212 | 0.61 | 0.69 | 0.27 | **0.76** | 0.6 | 0.7 | 0.66 |
| 10 | 204 | 0.56 | 0.64 | 0.26 | **0.7** | 0.55 | 0.63 | 0.58 |

Table 7: Performances of Different Models on MuseD over prompt levels.

| Levels | count | GPT-4 | GPT-4o | GPT-4-o1-mini | GPT-4-o1-preview | Qwen2-72B-Instruct | Qwen2.5-72B-Instruct | Llama3.1-72B-Instruct |
|---|---|---|---|---|---|---|---|---|
| 0 | 218 | 0.82 | **0.93** | 0.65 | 0.91 | 0.86 | 0.91 | 0.86 |
| 1 | 484 | 0.82 | **0.91** | 0.58 | 0.9 | 0.83 | **0.91** | 0.83 |
| 2 | 546 | 0.73 | 0.82 | 0.44 | **0.87** | 0.76 | 0.84 | 0.76 |
| 3 | 394 | 0.69 | 0.74 | 0.35 | **0.82** | 0.67 | 0.77 | 0.71 |
| 4 | 228 | 0.62 | 0.68 | 0.3 | **0.77** | 0.58 | 0.72 | 0.65 |
| 5 | 100 | 0.53 | 0.62 | 0.25 | **0.74** | 0.52 | 0.66 | 0.6 |
| 6 | 26 | 0.65 | 0.68 | 0.23 | **0.78** | 0.42 | 0.53 | 0.54 |
| 7 | 4 | 0.42 | 0.51 | 0.14 | 0.53 | 0.32 | **0.58** | 0.41 |

Table 8: Performances of Different Models on MuseD over noise counts.

As the result shows, LLMs become worse as the the level becomes larger. That is, the more complex the prompt is, the lower the step score is. GPT-4-o1-preview still keeps a 0.7 step score for level 10. Similarly, the more noise conditions are involved, the worse performances LLMs have.