# OpenReview forum: "Boosting Deductive Reasoning with Step Signals In RLHF"
_ICLR.cc/2025/Conference — Submitted to ICLR 2025_

### Official Review · Reviewer_2LpL · 2024-10-28

**Soundness:** 3
**Presentation:** 1
**Contribution:** 2
**Rating:** 3
**Confidence:** 4

**Summary:**

This paper introduces a new synthetic dataset, MuseD, consisting of multi-step deduction questions that are constructed from a novel tree generation algorithm that can be configured for varying levels of complexity. Because the questions are synthetic and based on syllogisms (and formal logic), the intermediate outputs can be evaluated from language models, which can then be given a score acting as a reward.  The authors use the intermediate scores from language models on the MuseD dataset to train reward models, followed by language models using PPO.  They show that by incorporating their dataset with dense positive rewards, LMs perform better on other deduction/reasoning tasks such as ProntoQA, FOLIO, and AR-LSAT.

**Strengths:**

1. Incorporating reward signals from other domains outside of math, I think, is exciting.
2. Including dense rewards for following specific steps does seem to have a positive impact on the language model after alignment

**Weaknesses:**

1. As of now, the paper is difficult to read and understand.  The mistakes in the grammar make many of the sentences and paragraphs difficult to understand what model/results/etc. are being discussed. Additionally, this paper would benefit greatly from figures explaining how the trees are constructed and another showing the types of questions with the reward structure broken down. Finally, I think there are so many models reported in Table 1, that using the subscript naming scheme is difficult to follow.  I would separate the ablated models like PPO_Fo-p and PPO_Na-R into their own sections, introduce them there and talk about them there rather than introduce all the models at once.  A figure for these or a table clearly showing how they differ from each other may also ease the burden on the reader.

2. There is no discussion on why models get questions from MuseD wrong.  This is especially important for this paper, I think, because it's a very toy setting.  I don't think "toy" is bad here, but I do find it surprising that o1 only gets 89% of these correct when they are questions like "all As are B. All Bs are C. Are all As C?" - from personal experience, I think it would take a very large set of premises for o1 / GPT-4 to start to fail, so I am curious why it's failing so often already.  I also think that general error analysis is always nice for papers like this (ones targeting behaviors in LLMs) so that researchers know where models fail on your dataset.

In short, the paper severely lacks clarity. I think fixing this, along with some error analysis, would greatly increase my score.

---

Less serious weakness: Section 3 takes up a page, and I think is mostly redundant.  The "middle term" is introduced here, but I think it'd be far more impactful to replace this section with an image and actually show the "middle term"s in an image to drive home the meaning.  I also do not think you should introduce notation like the A, E, I, O questions if they are not referenced again (didn't see it mentioned else where).  This is just another clarity thing.

**Questions:**

- Why do you need to train a reward model for this?  You could use an open-source one and use the gold truth reward function since the dataset is synthetic (as long as it produces text in a parseable format).

- What are the average sizes of the trees? I think I see a few trees at level 7 in the appendix from the supplementary material pdf, but average stats on the dataset would be nice.

- Did you explore performance across question types somewhere? You introduce the A, E, I, O questions from Aristotle, but I think I only saw "All X are Y" type of questions (or maybe they are all mixed together in your results)?

---

### Official Review · Reviewer_j8iL · 2024-10-28

**Soundness:** 2
**Presentation:** 3
**Contribution:** 1
**Rating:** 3
**Confidence:** 5

**Summary:**

This work proposes an automatic method, MuseD, to generate multi-step reasoning training/test datasets based on formal logic. After RLHF on the synthetic data, LLM achieves significant improvements in logical capabilities for logical reasoning tasks. Besides, the test dataset of MuseD can be used as a benchmark for LLMs.

**Strengths:**

This work proposes a simple but effective method to enhance the logical reasoning ability of LLMs. The performance of LLM (Llama3-8B) improves significantly on several logical reasoning tasks.

**Weaknesses:**

The contributions and experiments of this work do not seem solid.

Firstly, the methods and forms of the generated logical reasoning datasets seem overly simple, only reflecting multi-step features, and do not appear to be significantly different from previous works, like ProofWriter.

Secondly, the PPO-based model are only compared with the original baseline LLM (LLaMA3) and do not include comparisons with other baseline models. In fact, many fine-tuned smaller models have also achieved good performance on formal logical reasoning, such as ProofWriter.

Thirdly, the performance of the LLaMA model after secondary training seems to be inferior to that of the GPT models, this raises a question: is the data augmentation method provided in this paper equally effective on the GPT models or other LLMs?

Finally, from the results in Table 3, the statement "Among reasoning tasks, multi-step reasoning poses a particular challenge" doesn't seem to be true.

In conclusion, constructing formal logical reasoning datasets does not seem to be an innovative endeavor. Moreover, the formal reasoning capabilities of LLMs do not appear to be the primary challenge.

**Questions:**

(1) Can you provide more PPO results on other baseline LLMs and comparisons with other baselines introduced in the original paper of the evaluation tasks (ProntoQA, ProofWriter, LogicalDeduction, FOLIO, AR-LSAT). I believe that a relatively comprehensive presentation of the current state of AI in formal logical reasoning would help in understanding the value of this work.

(2) Why do you choose the RLHF strategies to train your model. The processes and results of logical reasoning themselves can provide the necessary knowledge for formal logical reasoning, without needing human experience as a supplement. Actually, many fine-tuning models can also perform formal reasoning without pre-training.

(3) What are the characteristics of the MuseD test set? From Table 3, LLMs seem to perform well on this dataset already.

---

### Official Review · Reviewer_oZPc · 2024-10-29

**Soundness:** 4
**Presentation:** 3
**Contribution:** 2
**Rating:** 5
**Confidence:** 3

**Summary:**

This paper explores logical reasoning of LLMs, specifically focusing on syllogism. The authors propose an automated approach for generating questions at various difficulty levels, along with step-by-step responses and a step-based scoring system. They use this score to fine-tune models with PPO, achieving better performance than baselines (untrained models and finetuned on a general dataset) and PPO models with only results reward. This scoring approach also offers a framework to evaluate current LLMs.

**Strengths:**

1. Deductive reasoning, especially syllogistic reasoning, is foundational for tackling more complex tasks. Fine-tuning on the proposed dataset meaningfully improves the model's ability to apply correct syllogistic reasoning.

2. Using a step-based signal for reinforcement learning is a reasonable approach. For tasks where sequential steps are crucial, step-level feedback can help the model learn accurate reasoning pathways more effectively during the RL process.

3. The experiments are thorough, including a detailed ablation study on various output formats (e.g., natural language, JSON) and different scoring compositions (step score, negative score, or result-only score).

**Weaknesses:**

1. While the use of step-level feedback or process-based rewards is intuitive, it is not novel and has been previously introduced by works such as [1] with subsequent advancements in [2, 3]. Automating label generation is crucial for training a reward model; however, since syllogistic reasoning is formal and symbolic, the potential step formats are highly constrained. Consequently, the step-level feedback here may be trivial, as identifying correct and relevant steps is straightforward.
2. The proposed automated labeling method is tailored specifically to the structured nature of syllogistic reasoning, limiting its applicability to other tasks. Additionally, models fine-tuned on this dataset appear sensitive to data shifts; in the OOD (out-of-distribution) experiments, PPO fine-tuning degrades performance on AR-LSAT, which involves a different logical paradigm. Also notice that the authors should add citation and introduction to these OOD datasets.
3. The OOD datasets ProntoQA, ProofWriter, and LogicalDeduction are quite similar to the in-domain syllogistic samples in the MuseD fine-tuning dataset. These datasets largely comprise syllogisms or syllogism combinations, and FOLIO also includes a subset of syllogisms. It would be insightful to test the fine-tuned models on a broader range of reasoning tasks to assess their generalization capabilities.
4. The hangling of those "incorrect steps" feels somewhat crude. Although these noise and irrelevant steps may not contribute to the correct answer, are they always a negative effect on the overall reasoning of the model? Are they necessary attempts at a reasonable reasoning process? Could they represent necessary exploratory attempts? It's worth questioning if a reasoning process that "goes straight to the correct answer" is indeed better --- or more aligned with human preference --- than one that includes reasonable yet unfruitful attempts. The experimental results suggest that penalizing incorrect steps can degrade performance, so a more nuanced discussion of these steps and their role in the reasoning process would add depth.

[1] Let's Verify Step by Step, Lightman et. al., 2023

[2] Math-Shepherd: Verify and Reinforce LLMs Step-by-step without Human Annotations, Wang et. al., 2024

[3] Let's reward step by step: Step-Level reward model as the Navigators for Reasoning, Ma et. al., 2023

**Questions:**

In addition to the points above, I have a few further questions:

1. How does the system verify that a step is both correct and relevant to the problem, particularly for responses in natural language? In Section 4.2, the authors mention that "natural language is relatively challenging to handle during scoring," but no details are provided on how this challenge is addressed.

2. In the judgment setting, conclusions are either correct or reversals of correct conclusions. However, does the dataset lack conclusions that cannot be determined (i.e., cases where neither the conclusion nor its reversal can be logically derived from the premises)?

3. The website referenced in the footnote is unavailable. Please check its accessibility. Additionally, it would be helpful to list the 15 formats in the appendix for reference.

---

### Official Review · Reviewer_mTKy · 2024-11-04

**Soundness:** 2
**Presentation:** 1
**Contribution:** 2
**Rating:** 3
**Confidence:** 3

**Summary:**

This paper introduces MuseD, a method for generating synthetic data of multi-step logical deductive reasoning for training LLMs. The authors focus on generating data for post-training (RLHF). MuseD can also serve as an evaluation benchmark, with finer-grained metrics on the quality of each step. Experiments show that RLHF on MuseD improves performance on OOD datasets, like PrOntoQA, ProofWriter, FOLIO, LogicalDeduction (a BigBench task), and AR-LSAT.

**Strengths:**

* The paper starts with a good premise, of generating synthetic reasoning data for training LLMs. Methods to generate high-quality synthetic data at scale are generally becoming more popular in the field, and I predict their importance to keep rising.
* The paper explores post-training, which is less explored in the reasoning space than fine-tuning approaches
* Results with Llama 8B seem to be mostly positive, and the authors compared to running RLHF on Ultrafeedback alone

**Weaknesses:**

* The paper doesn't show a single example of the data the method is able to generate (not even in the appendix). The explanation (Section 4) is a bit hard to follow, with details all only given in text. It would perhaps be more productive to discuss concrete examples, even if the details of the algorithm are discussed at a higher level (these can likely be inferred from seeing a few representative prompts). If I missed this, I'd appreciate if the authors point me to where such examples are.
* From the description, it seems like the problems in MuseD will look rather templated. It's unclear how the models generalize to much messier data like the problems in FOLIO, which are human-written and involve nuance in language and common sense (which the authors explicitly want to avoid; e.g L204 -- to avoid shortcuts).
* The authors should show that PPO is really necessary here, by trying an Supervised Fine-Tuning baseline on the same data. PPO is generally very unstable, and there's no a priori reason why it makes sense to focus on it.
* The many ways to train the reward model seem a bit ad hoc - I'm not sure I get the intuition behind many of the variations (Sec 5.2). Perhaps it would make sense to try something like DPO, with an implicit reward model, and thus less design choices to be made.
* Post-training experiments only done on Llama 8B

**Questions:**

* Why did the authors particularly focus on PPO for reasoning? For post-training, SFT and DPO [1] are often much simpler alternatives that often achieve comparable or better results. Did the authors consider these?
* Are there concrete examples in the paper of problems that MuseD is able to generate?
* What are all the rules of inference that MuseD uses?
* How much compute did the experiments in Table 1 use?

---

### Meta-Review · Area_Chair_SZoj · 2024-12-18

**Metareview:**

While the reviewers felt there were some merits in the paper (the premise of using logical reasoning inside LLMs), there were several areas of improvement as listed in the proposal. On reading the paper and reviews, I tend to agree with the reviewers that more work is needed but this paper does have potential after fixing the writing as clearly explained in the reviews.

**Additional Comments On Reviewer Discussion:**

Since the authors did not submit a rebuttal, the reviewers did not discuss this in depth.

---

### Decision · Program_Chairs · 2025-01-22

Reject